# Three-Dimensional (3D) Printing in Cancer Therapy and Diagnostics: Current Status and Future Perspectives

**DOI:** 10.3390/ph15060678

**Published:** 2022-05-27

**Authors:** Awaji Y. Safhi

**Affiliations:** Department of Pharmaceutics, Faculty of Pharmacy, Jazan University, Jazan 45142, Saudi Arabia; asafhi@jazanu.edu.sa

**Keywords:** 3D printing, Cancer-on-a-chip, bioink, drug screening, cancer detection

## Abstract

Three-dimensional (3D) printing is a technique where the products are printed layer-by-layer via a series of cross-sectional slices with the exact deposition of different cell types and biomaterials based on computer-aided design software. Three-dimensional printing can be divided into several approaches, such as extrusion-based printing, laser-induced forward transfer-based printing systems, and so on. Bio-ink is a crucial tool necessary for the fabrication of the 3D construct of living tissue in order to mimic the native tissue/cells using 3D printing technology. The formation of 3D software helps in the development of novel drug delivery systems with drug screening potential, as well as 3D constructs of tumor models. Additionally, several complex structures of inner tissues like stroma and channels of different sizes are printed through 3D printing techniques. Three-dimensional printing technology could also be used to develop therapy training simulators for educational purposes so that learners can practice complex surgical procedures. The fabrication of implantable medical devices using 3D printing technology with less risk of infections is receiving increased attention recently. A Cancer-on-a-chip is a microfluidic device that recreates tumor physiology and allows for a continuous supply of nutrients or therapeutic compounds. In this review, based on the recent literature, we have discussed various printing methods for 3D printing and types of bio-inks, and provided information on how 3D printing plays a crucial role in cancer management.

## 1. Introduction

Cancer is a major cause of death, with an estimated 19.3 million new cases and 10 million deaths in 2020 worldwide [1]. According to the Globocan Cancer observatory, breast cancer is the most common type of cancer, followed by lung cancer, colorectal cancer, prostate cancer, and stomach cancer. However, lung cancer is considered the foremost cause of cancer disease, followed by stomach, liver, and colorectal cancer [2]. Several cancer treatments are currently available for patients, such as surgery, radiotherapy, chemotherapy, immunotherapy, and hormone therapy. However, tumor heterogeneity increased treatment side effects and the recurrence of cancer, which makes cancer therapy more challenging for physicians and researchers—and thus, the search for the best strategies for cancer treatment is still increasing [3]. An important hurdle for the medical translation of effective anticancer medicines is due to an inconsistency in both in vitro and in vivo evaluations. Thus, there is an urgent need to develop physiologically relevant in vitro models to mimic the cancer environment to replace current drug development systems. 3D bioprinting is an emerging technique in which 3D structures can be developed using biomaterials containing cells and other biologically effective factors that are necessary to enhance cell growth and development, mimicking living tissue complexity [4].

Three-dimensional printing is recognized as additive manufacturing or rapid prototyping technology where the products are printed layer-by-layer via a series of cross-sectional slices with the exact deposition of multiple layers of different cell types and biomaterials based on the digital model designed using a computer used software. 3D printing involves four important steps: (1) 3D model generation using CAD software for X-ray, CT, and MRI; (2) bio-ink development using cells, growth factors, and hydrogels; (3) bioprinting using appropriate printing parameters; and (4) functionalizing the cells to generate tissue function through physical and chemical stimulation. The formation of 3D technology helps in the development of novel drug delivery systems through the development of innovative strategies [5]. 3D constructs of tumor models are developed using many cells, like normal and cancer cells, which can be deposited using 3D bioprinting, thereby obtaining microscale precision to mimic the tumor micro-environment and can be used as cell models for drug screening [6]. Additionally, several complex structures of inner tissues like stroma and channels of different sizes are printed using 3D printing technology [7]. Three-dimensional-printed models are used for surgery training, planning for the suitability of organ transplantation, planning of cancer-related surgery procedures, and training for performing surgery in small body cavities and complex delicate tissues [8,9].

Interestingly, in comparison with traditional pharmaceutical technologies, 3D printing technology effectively regulated the dose of tablets according to the patient’s needs by modifying the size or filling rate and helping in preparing individualized medicine [10]. Recently, orally disintegrating tablets printed by 3D printing technology are receiving a lot of attention due to the increased porosity and faster disintegration rate of the formulation [11]. Khaled et al. [12] stated the effective usage of printing technology in 3D to develop a “Polypill” consisting of 5 different active ingredients with effective personalized drug release behavior to the achieve a desired therapeutic effect and improve patient survival. Three-dimensional printing technology can also be used to develop a virtual model of a patient’s anatomy using a variety of imaging modals, with computed tomography and resonance imaging (MRI) leading to low-cost, high-fidelity anatomic simulations, which help in exact treatment delivery [13]. Thus, 3D printing technology can also be used to develop therapy training simulators for educational purposes so that learners can practice complex surgical procedures [14]. Apart from pharmaceutical drugs, nanomedicine is also fabricated by 3D printing technology as individualized medicine with surface functional modification together with bioconjugation and increased biocompatibility [15]. The fabrication of implantable medical devices using 3D printing technology with less risk of infection has been receiving increased attention recently. An experiment was carried out to test whether the use of 3D-printed objects loaded with gentamicin or methotrexate effectively decreased the growth of *E. coli* and suggested that antibiotic efficiency was not reduced while using 3D-printed objects [16]. In this review, we discussed various 3D printing methods, their importance, and the types of bio-inks. Importantly, we have given in-depth knowledge in understanding the role of 3D printing in cancer management like 3D printing in cancer surgery, drug toxicity/screening, Cancer-on-a chip, cancer detection, and cancer metastasis (Figure 1).

## 2. 3D Printing Methods

Three-dimensional printing technology helps in developing tumor models, the printing of various organs, producing implants, etc. In this section, we will be discussing printing methods that are widely used in medical fields. Based on different prototyping principles and printing materials, 3D printing can be classified into various approaches, like extrusion-based printing, inkjet-based printing, and the SLA-based printing laser-induced forward transfer-based printing system. Figure 2 below illustrates the 3D printing strategies and their application in the medical fields.

### 2.1. Extrusion-Based Printing

Extrusion-based bioprinting is amongst the most widely used methods for both biological and non-biological applications due to its affordable price and capacity to construct hollow/complex constructs [19]. It involves a temperature-controlled cartridge extruding the bio-ink from a fine nozzle with various forms of extrusion forces. Generally, the driving force can be a pressure-driven, pneumatic-driven, force and displacement-driven, or mechanical-driven force [20]. A study to generate organoids of the kidney with cell numbers having high reproduction and increased viability developed a 6- or 96-well organoid using extrusion-based 3D bioprinting. It enables the development of kidney tissue sheets in a uniform pattern with increased nephron cells together with a functional proximal tubular, and thereby effectively improves the quality control, scale, and structure when compared to the manual organoid method [21]. In another study, an extrusion-based bioprinting approach utilized bio-ink consisting of living cells, glucose, alginate with a phenolic hydroxyl group, and cellulose nanofiber to develop lattice and human nose-shaped 3D constructs that remained stable for more than a week [22]. Gospodinova et al. [23] reported that extrusion-based 3D printing helps in developing cervical cancer models by using hydroxyethyl cellulose-based bio-inks with different amounts of sodium alginate embedded with HeLa cells. The cell viability was not affected while using extrusion 3D printing and was maintained for a week. Moncal et al. [24] established a new bio-ink containing Pluronic^®^ F-127 hydrogels and collagen type-I and printed using extrusion-based 3D printing. The collagen fibers align themselves in the direction of the printed filaments. Additionally, to study the influence of the thermally controlled extrusion process, the rat bone marrow-derived stem cells (rBMSCs) were bioprinted. The investigation suggested that the cell viability was stable and the cells were attached to collagen fiber to proliferate and migrate, and the cell culture was stable for a week. The graded biomaterials using bio-ink carboxylated agarose were printed by extrusion-based 3D bioprinting, which prints an object with a gradient of stiffness and cell concentration. Thus, it could help develop tissue biomimetics with gradients [25].

An extrusion-based 3D bioprinting process was used for fabricating cell-laden constructs using human mesenchymal stromal cells and silk fibroin–gelatin (G)-based hydrogel and tested for the conversion of stromal cells into chondrogenic cells. The data showed that the cells developed a stable chondrogenic phenotype by expressing collagen and filamin b without increasing cell viability [26]. Calcium chloride, a preprint cross-linking agent, was added into native nano-fibrillated cellulose hydrogel before extrusion-based 3D printing, which helps fabricate stable multilayered constructs without using a separate cross-linking bath. Then, the mouse embryonic fibroblast cells were cultured into hydrogels and the authors found that calcium chloride enhanced the stability of 3D printed structures and improved the viability of the cells [27]. In a study, the development of an artificial ovary was studied using extrusion-based 3D bioprinting. However, they reported that the viability of cells was lower in extrusion-based 3D culture when compared to commercial cell lines and suggested that extrusion-based culture fabrication is not suitable for the development of the artificial ovary. Alternatively, the gelatin-methacryloyl-based 3D printing system provided the necessary environment for the growth of ovarian follicles in the scaffold, and thus it could be used as an alternative strategy for follicular growth and used for the treatment of female reproductive conditions [28]. Human hepatoma cells and mouse fibroblasts bounded in the printed extrusion-based 3D model and exhibited an increased viability of about 95% on the next day of printing, which remained stable for 11 days. Thus, the study suggested that phenol-grafted polyglucuronic acid can be used in the field of tissue engineering, particularly as an ink component of extrusion-based 3D bioprinting [29] (Figure 3).

### 2.2. Laser-Assisted 3D Printing System

Laser-assisted 3D bioprinting is a kind of bioprinting technique via laser energy as the basis for the development of tissues designed artificially, and this 3D printing is created according to the laser-induced forward-transfer (LIFT) effect, which helps in diverse biomaterials and living cell deposition to generate scaffold-free 3D cell systems through a layer-by-layer process to form stable gels [31]. It is made up of three important types of machinery: (i) a pulsed laser source, (ii) a target to print biological material, and (iii) a substrate as a receiver to collect printed material. The setup of laser-assisted bioprinting is a nozzle-free system and involves the coupling of a near-infrared pulsed laser to a focus system using a scanning mirror, which would focus the laser beam towards the biological material [32]. Based on the CAD modeling, laser pulses are focused on the target area to generate a high-pressure vapor pocket, leading to the formation of a cell-laden droplet, which would drop on the receiving substrate and cross-links. This technology prevents cell clogging and does not affect cell viability. Due to its increased high throughput capability and reproducibility, it can be used to generate 3D-printed pre-cancerous and cancer models [32]. Hakobyan et al. [33] developed reproducible 3D cellular spheroids arrays consisting of acinar and or ductal exocrine pancreas cells using laser-assisted bioprinting and could be used as a 3D model to study the development of the initial stages of pancreatic ductal adenocarcinoma.

Laser-assisted bioprinting (LAB) was used for developing corneal tissue mimics via human embryonic stem cell (hESC)-derived limbal epithelial stem cells, lamellar corneal stroma with alternating acellular layers of bio-ink, and layers with human adipose tissue-derived stem cells (ADSCs). After printing, the 3D constructs exhibited good viability for the adipose stem cells and the epithelial cells were organized similar to native corneal stroma with migration potential. Ultimately, the study showed that they had successfully developed layered 3D bioprinted tissues mimicking corneal tissue [34]. In another study, it was revealed that LAB helps in printing and organizing nano-hydroxyapatite and human osteoprogenitors without changing viability and proliferation for up to 15 days. Also, it was suggested as a significant technique for patterning cells in two dimensions and is involved in the fabrication of 3D composite materials [35]. Tissue engineering plays a crucial part in the treatment of chronic skin conditions and burn wounds. A study has utilized the LAB technique to develop fully cellularized skin as it could effectively use different cell types in 3D spatial patterns. They used fibroblasts, keratinocytes, and Matriderm^®^ to develop skin substitutes and tested them in in vivo nude mice. The data showed that cells had undergone differentiation and proliferation, which leads to tissue formation and, interestingly, small blood vessels were grown towards printed tissue from the wound bed and edges [36]. In another study, ADSCs were fabricated into a pattern of a 3D grid using LAB and it was found that cell proliferation and differentiation were not affected after printing. The adipogenic marker expression revealed that the cell lineages resemble 3D grafts similar to that of natural adipose tissue [37]. A study has developed injectable micro-scaffolds from electro spun material using LAB, which has the capacity to produce ten thousand micro-scaffolds within a short period of time with a high injectability rate. Additionally, the cells were populated on micro-scaffolds, and it was found that micro-scaffolds act as cell carriers and can be used to study minimally invasive cell therapies in more depth [38].

### 2.3. SLA-Based Printing

Stereolithography (SLA) printing was the technique reported initially to develop 3D complex constructs with more accuracy and resolution and has two important factors compared to LAB methods and extrusion-3D printing. (1) Objects can be made at room temperature. (2) The degradation of the drug can be avoided by drug incorporation into resin [39]. Similar to LAB, SLA also uses light sources ranging from ultraviolet to visible light to cross-link or polymerize the bio-ink for the development of 3D constructs. There are two types of polymerizations: The first is image projection and the second is beam-scanning. The image projection method uses 2D images, while beam scanning involves polymerization along with drawing patterns of light [40]. SLA has been used to develop in-dwelling bladder devices that are solid and hollow and are made up of lidocaine hydrochloride in a three-drug load quantity, which can be inserted into and removed from the bladder through a urethral catheter. The devices showed good biocompatibility with lidocaine release from hollow devices within 4 days, while solid devices showed drug release for 14 days [41]. A comparative analysis was shown to understand the material properties of polyethyleneglycoldimethacrylate (PEGDMA) between UV chamber photopolymerization and SLA 3D printing. Compared to SLA 3D printing, it exhibited higher compressive and tensile strength with enhanced hydrophilicity [42]. In a study, personalized medicine was fabricated using SLA printing by biocompatible photochemistry using ascorbic acid encapsulated in a PEGDMA-based polymer network and polymerized using photoinitiator riboflavin. Based on the gastrointestinal release rate, the data showed that the tablet microstructures of a honeycomb and coaxial annulus showed a higher release rate of up to 80%, which ultimately suggested that this method could be effectively used for drug delivery applications [39]. Similarly, in another study, the SLA-printed-drug-releasing potential was analyzed using PEGDA, photoinitiators, and model drugs (4-aminosalicylic acid and paracetamol). The gastrointestinal release rate suggested that SLA 3D technology could help in the manufacture of drug-loaded tablets with specific extended-drug release potential [43]. In order to train surgeons and test new medical devices, synthetic bone models will be developed. In a study, SLA printing was used to develop synthetic trabecular bone based on micro-CT images and found that SLA-printed bone parts showed a higher pull-out strength compared to existing synthetic SawbonesTM with higher resolution [44].

A transdermal microneedle that is 3D printed was fabricated for insulin delivery using SLA and found that skin penetration capacity was higher in 3D printed microneedles with minimum applied force when compared to metal arrays. Additionally, in vivo studies showed that glucose is lowered within 60 min due to fast insulin action with steady-state plasma glucose in transdermal injection when compared to subcutaneous injection [45]. Decellularized tendon extracellular matrix (tECM) is essential for bone regeneration and, in a study, tECM and PEGDA scaffolds with an appropriate pore size and strength were fabricated using SLA and suggested that 3D printed polyporous PEGDA/tECM (3D-pPES) scaffolds can be effectively used for bone defect treatment based on the data, which showed an increased cell migration potential, enhanced osteogenic differentiation, and effective calvarial defect repair capacity in a rat model of 3D-pPES when compared to the control [46].

### 2.4. Inkjet-Based Printing

Inkjet bioprinting is widely recognized as the first bioprinting technology, followed by extrusion-based printing [47]. The inkjet printing process involves two important steps: (1) the formation of the droplet and directed toward location of substrate and (2) droplet and substrate get to interact. Continuous inkjet printing and drop-on-demand inkjet printing are the two types of inkjet printing. Drop-on-demand inkjet has a higher printing resolution with a lower drop generating frequency when compared to continuous inkjet, which has higher drop generating frequencies and possesses sterility issues [48]. Corneal opacities are an important cause of blindness and treatment strategies involve the use of a donor cornea. In a study, polymer hydrogel was developed using reactive inkjet printing and used corneal epithelial and endothelial cells to attach to the surface of printed hydrogels, and thus was involved in fabricating the corneal construct [49]. In a study, matrix material (Compritol and model drug, Fenofibrate) were used to prepare loaded inks and were drug-free to develop personalized printed dosage forms using inkjet printing. Compritol was printed using hot-melt inkjet printing either in combination with a drug or single ink material to produce multi-material personalized solid dosage forms. The printed constructs demonstrated that drug release completely depends on the localization of the drug inside the printed formulation [50]. Water-based ink preparation was established using polyvinylpyrrolidone and thiamine hydrochloride (model drug) and the tablets were printed on polyethylene terephthalate (PET) films using inkjet printing. The printed tablet showed a rapid drug release, and the use of solvent can be avoided; thus, this strategy of printing helps in preparing water soluble-drug formulations [51].

Piezo-activated inkjet 3D printing was used for producing tablets with PEGDA hydrogel matrix containing ropinirole hydrochloride and photoinitiated using aqueous Irgacure 2959, and, ultimately, this strategy showed an increased drug release [52]. Irgacure 2959, a photocurable N-vinyl-2-pyrrolidone (NVP), and PEGDA were used in 3D inkjet printing with UV curing to develop solid dosage forms containing the poorly soluble drug, carvedilol. Eighty-percent carvedilol was released within 10 h from the printed tablet and the release rate depends on the tablet geometries showing an increased release for thin films followed by the ring and meshes, while the slowest was observed in cylindrical geometry [53]. In that other research, hot-melt 3D inkjet printing has been used to generate a formulation for controlled drug release. They have reported that FDA-approved beeswax was used as a drug carrier (fenofibrate carrier) and a honeycomb architecture was fabricated. Surface area, cell size, and material wettability must be considered for designing a formulation, and this strategy can be optimized for personalized medicine and can be used for the delivery of various tablets [54]. A combination of spray-coating upon drop-on-demand inkjet printing was used for fabricating hydrogel structures using different materials like fibrinogen, cellulose nanofiber, and alginate. By evaluating the microstructure and mechanical stiffness, cell viability, and function of human dermal fibroblasts in hydrogels, the data showed that the inkjet-spray printing method will help in fabricating laden hydrogel structures with high fidelity and can be applied for 3D laminated large-scale tissue equivalents that mimic native tissue [55].

Due to the outbreak of a new respiratory virus and its associated respiratory disease, there is an increased need for the development of a respiratory model to study disease pathogenesis. Using drop-on-demand inkjet printing, a 3D three-layered alveolar barrier model was developed using type I and II alveolar cells, lung fibroblasts, and microvascular cells. The results showed a better structure, morphologies, and functions of the lung tissue than a 2D model. The study suggested that the 3D alveolar barrier model can be used as an alternative tool to be used for pathological and pharmaceutical applications [56]. Similarly, in another study, drop-on-demand inkjet printing could be used for printing mesenchymal, stromal, and chondrocyte cells in a custom agitation system, which would prevent the agglomeration and sedimentation of cells during printing. Additionally, a cell assay revealed that the agitation process didn’t affect the cell function, morphology, and viability of mesenchymal, stromal, and chondrocyte cells [57].

## 3. Bio-Ink

A 3D construct of living tissue is fabricated using bio-ink in order to mimic the native tissue/cells using 3D printing technology. In this section, we have discussed the various bio-inks and their role in the 3D printing process. A bio-ink consists of cells together with a natural/synthetic polymer matrix (gel) and the purpose of the gel is: (1) to act as a platform for the cells to adhere, increase, and differentiate; (2) to cross-link for developing a desired construct; and (3) prevent cell damage during printing [58]. Based on the printing method, bio-ink should possess tunable, mechanical strength and viscosity to support the growth, viability, proliferation, and functionalization of cells. Bio-inks are categorized into: (1) Protein/peptide polymer-based bio-ink (2) Carbohydrate-based bio-ink, (3) Extracellular-based bio-ink, (4) Synthetic polymer bio-ink, (5) Cell aggregate bio-ink, and (6) Composite bio-ink, which can be used for regenerative medicine, drug delivery, and tissue engineering [59]. In this section, we have discussed the most widely used bio-inks for fabricating 3D constructs. Table 1 shows the use of bio-inks for various biomedical applications.

### 3.1. Protein/Peptide Polymer-Based Bio-Ink

Protein/peptide polymer-based bio-inks include collagen, collagen mimetic peptide, helix-loop-helix-polypeptide, RGD peptide, fibrin, and gelatin. Collagen is one of the extracellular matrix molecules and is widely used as a hydrogel for the preparation of bio-ink. Based on the inspiration from animal tendon, a study has developed a pre-oriented bio-ink using collagen liquid crystals that possess excellent wet strength and exhibit wound suture capacity with an increased biodegradability potential, thus suggesting that it could be used for a variety of uses in the biomedical industry [69]. In another study, 4% collagen and chondrocyte were used as a bio-ink for the formation of cartilage. For that, extrusion-based bioprinting was used to print cartilage and showed that cartilage formation was found within a week with groups of isogenic cells and higher expression of glucosaminoglycan and type II collagen [70]. A study has designed a helix-loop-helix peptide conjugated with a hyaluronan backbone to enable cross-linking and hydrogel functionality, thereby exhibiting a modular and tunable 3D printable hydrogel system that can also be suitable for 4D printing [71]. Finally, it was proposed that laden -TCP is required for the osteogenic differentiation of hASCs, and thus collagen with bioceramide-based bio-ink could be effectively used for bone tissue regeneration [72]. A study has developed synthetic self-assembling peptide hydrogels, which would entrap 99.9% of water and mimic native collagen. Based on mouse myoblast cell viability and cytotoxicity studies, as well as their printability effect, it was shown that peptide hydrogels are biocompatible and suitable for developing 3D bioprinted scaffolds with skeletal muscle cells and could be used for tissue engineering [73].

A bio-ink containing novel RGD peptide altered gellan gum together with primary cortical neurons was used to fabricate a 3D-brain like structure. Peptide modification helps in effective proliferation and neuronal cell network formation, and thus the developed brain-like structure could be effectively used for conducting research and understanding brain damage and neurodegenerative ailments [74]. Fibrin is formed from fibrinogen and is necessary for blood clotting. ADSCs were bioprinted using a fibrin-based bio-ink and the cells were then treated with various factors, such as brain-derived neurotrophic and fibroblast growth factors to induce cell differentiation to form dopaminergic neurons, which would express neuronal markers after 12 days. Ultimately, the study showed that bioprinted, patient-derived mesenchymal stem cells could be used for developing neural tissues and could act as an important strategy for the development of a personalized disease model [62]. Similarly, in another study, a fibrin-based bio-ink preparation with drug releasing microspheres (guggulsterone) was developed using human-induced pluripotent stem cells derived from neural progenitor cells. The stem cells were differentiated into dopaminergic neurons and exhibited 95% cell viability after a week of printing. The printed tissue exhibited a dopamine marker, oligodendrocyte progenitor marker, and glial marker after 30 days, which ultimately suggested that microsphere-laden-based bio-ink can be used to promote the differentiation of neural tissue [75]. Bio-ink was developed using fibrin together with photo-polymerizable gelatin, which was used to develop cardiac cell-laden constructs using human iPS-derived cardiomyocytes or cardiomyocyte cell lines with fibroblasts. The constructs were then cross-linked by combining a visible light cross-linking of furfuryl-gelatin and a chemical cross-linking of fibrinogen with thrombin and CaCl_2_, thereby resulting in a porous networked structure. Thus, this construct was suggested to be used as a model to analyze drug screening and also to understand heart diseases [76]. Bioactive nanoparticles were established to release silicon ions and also used to stimulate the potential of alginate/gelatin hydrogel bio-inks in order to maintain the stemness of MSCs and support the growth of mesenchymal stem cells, thereby providing a new strategy for developing the therapeutic potential of stem cells in bioprinting applications [77]. A bio-ink was prepared using gelatin, carboxymethyl cellulose, and alginate for developing a 3D scaffold with osteosarcoma cells and it was observed that the enhanced collagen secretion, cellular proliferation, and biocompatibility effectively made the scaffold suitable for cartilage tissue engineering uses [78]. A hybrid bio-ink was developed using gelatin, carbon nanotubes, and sodium alginate to prepare cylindrical scaffolds that were inoculated with epidermal fibroblasts to fabricate blood vessels. It has been suggested that the doping of carbon nanotubes shows very little cytotoxicity and that the constructs can fulfill the criteria of biomimetic vascular systems [79].

### 3.2. Carbohydrates-Based Bio-Ink

Several carbohydrates-based bio-inks have been developed recently and used for biomedical applications, including cellulose, alginate, agarose, and hyaluronan. A bio-ink comprised of alginate together with boronic-acid-functionalized laminarin was used for 3D constructs, which improved the mechanical parameters due to cross-linking chemistry, stability, and tenability. Osteoblast precursors, fibroblasts, and breast cancer cells were printed using the bio-ink and found that cell viability was more than 90% and could be maintained until 14 days post-printing, which suggested that it could be used for various biomedical platforms [80]. A bio-ink formulation comprising of alginate, gelatin, and diethylaminoethyl cellulose was used to fabricate skin tissue equivalents. The bio-ink was reported to be non-cytotoxic and stable, and, upon loading with cells like human fibroblast and keratinocyte, the 3D constructs exhibited increased cell viability, enhanced collagen expression, and other skin-specific markers. Ultimately, it was suggested to be used for the growth of skin tissue equals with separate epidermal–dermal histological characteristics [81]. Hyaluronan-based bio-ink was characterized using two cross-linking mechanisms: (1) enzymatic method to form a soft gel that would be suitable for cell encapsulation and (2) visible light photo-cross-linking to shape 3D constructs. Stem cells, chondrocytes, and fibroblasts were encapsulated, and viability was observed for 14 days post-printing and could be effectively used for producing 3D tissue-engineered constructs. Considering the in vitro production of small- and large-sized vessels, alginate di-aldehyde and gelatin have been used for the fabrication of vessel structures with the size of 4 mm diameter to support the fibroblast and endothelial cell proliferation and migration. Based on the cell viability, it was suggested that it could be suitably used for the bio-fabrication of the vessel, like 3D constructs [82]. A combination of hyaluronic acid (HA) and collagen I bio-ink was used for extrusion bioprinting to support native cell–matrix interactions and preserve the native microenvironment. For that, primary human hepatocytes and liver stellate cells were used to bioprint 3D liver constructs and were tested using a liver toxicant acetaminophen. The data showed that cells persisted after printing for 2 weeks and also responded to drug treatment [83]. Hyaluronic acid, the main cartilage component, was used as a new bio-ink together with polylactic acid for the fabrication of cartilage tissue 3D constructs. The bio-ink effectively improved cell functions by showing an increased expression of chondrogenic gene markers, matrix deposition, and ultimately led to cartilage tissue formation, and thus was suggested to be a promising bio-ink candidate for cartilage tissue engineering 3D constructs [84]. Micro-extrusion-based bioprinting techniques employed a hybrid bio-ink using alginate, cellulose nanocrystal, and gelatin methacryloyl for printing cell-laden and acellular structures. Liver lobule-mimetic 3D constructs were bioprinted using NIH/3T3 and hepG2 cells embedded using a hybrid bio-ink. Enhanced hepatic cell function was observed due to increased albumin production and, ultimately, the data suggested that complex constructs with many cell types can be bioprinted and could be used for biomedical research and tissue engineering uses [85].

### 3.3. Decellularized Extracellular Based Bioink

To provide cells with a natural micro-environment, decellularized extracellular matrix (dECM) is a suitable method for a natural or synthetic material to recapitulate all the characteristics of normal ECM, which can act as an interaction between cells and the micro-environment [86]. Recently, dECM has been used to provide the necessary environment to fabricate 3D tissue constructs with the advantage of providing enhanced cell survival and function. In this section, we will be discussing the use of dECM in the biofabrication of various tissue 3D constructs. dECM pre-gel was used to bioprint hASCs or hTMSCs and the data showed that cell survival and proliferation remained unchanged, while a surprising increase in the adipogenic lineage and chondrogenic lineage were observed from the stem cell transition. Thus, this strategy could be used to improve physiological or pathological in vitro 3D models to conduct research on drug screening and toxicology studies with dECM, which would offer a precise tissue-mimicking 3D environment [87]. A cross-linker-free bio-ink was developed using cartilage ECM, silk fibroin, and bone marrow MSCs for 3D printing. Silk fibroin and dECM interact with each other through a physical cross-linking and porous structure, which was designed upon removing PEG from the bio-ink and provides suitable mechanical strength and helps in the increased expression of chondrogenesis-specific genes compared to that of a silk fibroin control construct. Finally, it was reported to provide a good cartilage repair environment and could be considered as an effective scaffold for cartilage tissue engineering [18]. Decellularized kidney ECM was used for a bio-ink preparation to 3D print renal progenitor cells and the enzymatic cross-linking of the dECM was done using transglutaminase. The encapsulation of primary renal progenitor cells showed improved cell viability, growth, and differentiation. Ultimately, the authors proposed that this method provided a good printing resolution with increased structural integrity and could be used as an effective application for in vitro model systems [88].

Three-dimensional biomimetic vagina tissue printing was performed using a cellular vagina matrix bio-ink and encapsulated bone marrow MSCs. For that, 15% gelatin and 3% sodium alginate were combined with the matrix, and, after the printing, the 3D scaffold with cells developed vascularization and epithelization—and thereby acquired vaginal epithelial cell and endothelial cell phenotypes. The study suggested that biomimetic 3D vagina tissue can be developed using an acellular vagina matrix encapsulating BMSC and it could also be used for vagina reconstruction [89]. A study has developed a two-step process to enhance the mechanical properties of 3D constructs, which involves vitamin B2-induced ultraviolet A cross-linking and the solidification of dECM using thermal gelation. This combination mimics the native micro-environment of the heart tissue and it supported high viability, the proliferation of cardiac progenitor cells, and enhanced differentiation into cardiomyogenic cells, and thus provided a new approach for dECM-based 3D printing [90]. Liver dECM bio-ink was developed and studied for stem cell differentiation and HepG2 cell functions and compared with commercial collagen bio-ink, which was also evaluated, and the report showed that, comparatively, the liver dECM bio-ink effectively induced stem cell differentiation and increased HepG2 cell function, which suggests it could be an ideal bio-ink candidate for 3D constructs for liver tissue engineering [91]. Cartilage-derived dECM (cdECM) was administered into a photo-cross-linkable hydrogel utilizing methacrylate, and then chondrocytes were added to form a printable bio-ink. Then, the bio-ink was printed in an anatomical ear shape, while the viability and proliferation of auricular chondrocytes were in the printed cdECMMA hydrogel and produced cartilage collagen and glycosaminoglycans; this method could be an alternative strategy for auricular cartilage reconstruction [92]. Similarly, in another study, the digital light process-based bioprinting was performed using liver microtissue and liver dECM while the viability and proliferation of hiHep cells were maintained with better liver-specific functions like secreting albumin and urea. Thus, it was reported that liver dECM-based cell-laden bio-ink for liver microtissue fabrication could be used for liver tissue engineering [93]. A study has developed a bio-ink consisting of decellularized porcine myocardial extracellular matrix and reduced graphene oxide to provide a micro-environment for the growth and development of hiPSC-derived cardiomyocytes [94]. In another study, dECM is used for the development of 3D head and neck in vitro tumor constructs [95].

### 3.4. Synthetic Polymer Bioink

Synthetic polymer bio-ink is easier to regulate for cross-linking, mechanical strength, and high tunability. The most commonly used synthetic polymer bio-ink used for 3D printing contains poly(ethylene glycol)-tetraacrylate (PEGTA), poly(ethylene glycol) diacrylate (PEGDA), poly lactic-co-glycolic acid, and poly(caprolactone). In a study, PEGDA and silk methacrylate were used to develop photo-cross-linkable bio-ink together with chondrocytes for the biofabrication of 3D bioprinted cartilage constructs, which increased the expression of aggrecan and collagen type II and suggested that silk methacrylate (SilMA)-polyethylene glycol diacrylate (PEGDA) bio-ink could be a potential candidate for bioprinting chondrocytes to develop cartilage tissue repair and regeneration [96]. In a study, it was reported that composite hydrogel containing 30% PEGDA-7% GelMA/0.1% brilliant black was used to print a hollow vascular network, and, after printing human umbilical vein endothelial cells, showed an increased survival rate one week post-printing and exhibited effective biocompatibility of the composite hydrogel [97]. In another study, PEG-clay nanocomposite cross-linking hydrogel has been developed to fabricate the 3D printing of osteoblast cells. The encapsulated osteoblast showed increased viability and undergone differentiation into the osteogenic cell, and this strategy helps in bone tissue regeneration [98].

Bio-ink was developed using PPy-block-poly(caprolactone) [PPy-b-PCL]) for the bioprinting of neural tissue constructs. Based on cell viability, it was reported that the printability of collagen/PPy-poly(ε-caprolactone)-b(b-PCL) hydrogels exhibited good printability and cells, which showed enhanced viability and, ultimately, they suggested that this method can be effectively used to construct a study to repair the mechanism for neuronal tissue damage and for drug screening [99]. PCL was used to print a meniscus construct, and it was mixed with polyurethane and cell-laden dECM to develop bio-ink with the potential of high controllability and durability. The construct helps grow, proliferate, and differentiate the encapsulated stem cells toward fibro-chondrogenic and the construct exhibited biocompatibility, good mechanical strength, and increased functionality, and thus suggested that it could be effectively used for clinical practice [100]. A biomimetic polycaprolactone/hydrogel composite scaffold was developed by encapsulating chemokines and chondro-inductive molecules and also promoted stem cell horning and ketogenisis in poly(lactic-co-glycolic) acid microspheres [101]. In a study, NIH3T3cell-laden hydrogels and collagen-coated poly (lactic-co-glycolic acid) (PLGA) scaffolds were developed for the multiple cells, and data suggested that this method facilitated the direct spatial organization and hierarchal 3D assembly of multiple cells and could be used for tissue engineering uses [102]. Polyvinyl alcohol (PVA), a thermoplastic synthetic polymer with good solubility in water and good mechanical properties, showed a transition temperature of 85 °C and was used to produce multilayers of the polymer for additive manufacturing by the inkjet 3D printing method [103,104]. Poly(lactic acid) is a biodegradable polymer melting point of 150–175 °C and a Tg of 55 °C with poor solubility in water and increased solubility in solvents like dioxane, acetonitrile, and dichloroacetic acid, and has been successfully used in medical devices developed using various 3D printing techniques [105,106].

## 4. 3D Printing in Cancer Management

Cancer is one of the world’s top causes of mortality; despite various treatment strategies, there are still many challenges with the current cancer management approaches related to surgery, drug development, in vitro and in vivo models, and diagnosis methods [107]. Considering the above limitations, 3D printing technologies provide an excellent platform to offer a better solution for cancer management. It improves cancer surgery by helping in pre-surgical planning and acts as a teaching tool for patients and trainees and enhances personalized cancer therapy [108]. In this part, we will be discussing the use of 3D printing techniques for cancer therapy and diagnostics (Table 2) (Figure 4).

### 4.1. Cancer Surgery and 3D Printing: Clinical Studies

In cancer surgery, new surgical instruments with rapid advancements have been developed. Still, it is challenging to succeed with the required surgical methods and perioperative management. Recently, the utility of 3D printing has aided surgeons in planning for surgery, reducing the risk duration of surgery, increasing the therapeutic effect, and discussed the importance of 3D printing for the surgery of various cancers [118]. In a study, 61 patients with right hemicolon cancer who had undergone laparoscopic surgery were categorized into 3 groups: control (n = 22), 3D printing (n = 20), 3D-image (n = 19), and were analyzed for the importance of 3D printing in surgery. Comparatively, it has been reported that 3D printing effectively reduced the surgery time, bleeding volume, and number of lymph node dissections, and thus it could be more useful for novice surgeons [119]. Patients who have undergone malignant pelvic bone cancer surgery (n = 12) via a 3D-printed bone-cutting guide and reconstruction with a 3D-printed implant were analyzed using clinical information. Three-dimensional printed guides helped patients in such a way that pathologic marines were negative with faster rehabilitation. Additionally, it was reported that compared to the anatomical filling of bone defects, 3D printed implants could be reconstructed and used [120].

In a study, patients’ CT data were collected for 3D reconstruction and 3D printing, and 3D models were developed and used for understanding the association between a tumor and the hepatic bile duct, artery, portal vein, and hepatic vein in order to plan for surgery and perform simulated surgery. The data showed that liver failure or patient death was not observed perioperatively and, ultimately, they suggested that 3D printing helps for surgical safety and reduces surgical risk [121]. Ten of the twenty patients diagnosed with macroadenoma and undergoing endoscopic transsphenoidal surgery used 3D printing technology. The clinical results showed that patients who received 3D technology experienced less surgical time, reduced complication rate, and, altogether, it helps patients to experience a good prognosis [122]. In another study, it was concluded that Colorectal surgery could be performed by using 3D printing technology by improving patient education before stoma construction and helps in pre-operative surgical planning and the investigation of liver metastasis to chemotherapy via 3D ultrasonography [123]. Hong et al. [124] described that explaining the thyroid gland and its structure and surgery is very complex for the clinicians to communicate with the patients. In a study, a 3D-printed thyroid gland with cancer was obtained from CT of patients and the phantom showed the complex structure of veins, arteries, nerves, and other organs surrounding cancer. Thus, this technology helps clinicians to educate patients and helps in understanding the disease. A hybrid 3D model simulating laparoscopic choledochal surgery was developed using a 3D systems project 660pro with visit pxl core powder. However, the study showed that further development is necessary for this choledochal cyst excision simulation [125]. The pre-operative imaging of sixteen patients was investigated and the lung hilum was 3D printed in three patients. Three-dimensional-printed hilum was observed to be more accurate than 3D-reconstructed CT, and thereby suggested that 3D printing is necessary for planning for thoracic surgery and is more beneficial than conventional imaging modalities [126]. In a study, the possibility of radioactive 125I seed (RIS)implantation guided by CT together with non-coplanar template 3D printing in 66 locally recurrent rectal cancer patients were investigated and found that this method is a very effective treatment strategy for patients after surgery or external beam radiotherapy [127]. A randomized clinical trial was conducted for improving patient knowledge in informed consent to use personalized 3D-printed models for lung cancer (Stage I) surgery. Based on scores of patient knowledge, advantage, disadvantage, alternative treatments, and satisfaction, it was observed that personalized 3D printing could be used in patients suspected of possessing lung cancer (stage I) [128]. As data procured from CT machines could not be printed without processing, using CT image source data for 3D printing is reported to be highly desirable. A cost-effective 3D printed skull was developed with the structure of the nasal cavity at various stages of pituitary cancer and this printed model could be used for the surgery preparation of the endonasal trans-sphenoidal pituitary. Additionally, neurosurgeons and medical students can also practice surgery with different stages of tumors [129]. A study was conducted among the lung cancer patients, and they were categorized into different groups based on 3D chest CT reconstruction, 3D printing, and chest CT scans with image enhancement for 3D reconstruction. Based on operation time, blood loss, and post-operative complications, it was observed that comparative 3D printing methods help the exact location of nodules and can improve surgical safety [130]. In a study, a 3D-printed model with a skull base, cerebral arteries, and tumor/aneurysm was developed using a 3D model and 49 simulated surgeries were conducted under a microscope and an actual surgery was performed after getting experience. The authors suggested that 3D-printed craniocerebral models effectively provide simulated surgery conditions and help in surgical plans, experience, and validation in actual surgery [131]. The classification of medical bioprinting and its application has been presented in Figure 5.

### 4.2. Tumor Microenvironment and 3D Printing

The tumor micro-environment comprises cells like fibroblasts, epithelial cells, stroma, blood vessels, immune cells, signaling molecules of both tumor and normal cells, and ECM [133]. In spite of 3D co-culture and microfluidic systems, many challenges still remain in developing tumor micro-environments. Recently, 3D bioprinting has emerged as a novel method for the fabrication of complex tissue models with various biomedical applications, and we discussed the role of 3D printing in the tumor micro-environment. In a study, 3D glioma stem cells showed a greater potential to form spheroids, develop tubule-like structures, secrete VEGFA, and also effectively differentiate into endothelial cells. Additionally, it was reported that a 3D bioprinted hydrogel scaffold helps in providing the necessary tumor micro-environment for glioma cells and GSCs [134]. Acoustic droplet 3D printing doesn’t have a nozzle and doesn’t get clogged. It increases cell viability and increases the number of cancer-associated fibroblasts in the tumor micro-environment, which leads to functional native tissue or diseased models [135]. In a study involving tumors and fibroblasts for a developing tumor micro-environment with a microfluidic channel, a 3D printed plastic brick-like microfluidic gadget was fabricated, which effectively involves heterotypic co-culturing and aids in phenotype decoding and molecular assays. The data obtained were validated using a mouse xenograft model and it was found that the 3D in vitro method helps in the understanding of tumorigenesis and the associated tumor micro-environment [136]. A 3D bioprinted GelMA/PEGDA hybrid scaffold mimicked the tumor micro-environment of human malignant melanoma cell and was reported to be suitable for the expansion and differentiation of tumor cells; additionally, tumor cells were growing faster and exhibited drug-resistant potential [137]. Alginate and gelatin bioprintable hydrogel together with BC cells and fibroblast were printed to form a 3D mimicking tumor micro-environment. This approach amplified the viability of cells and enhanced the development of tumor spheroids, which would interact with cancer-associated fibroblasts, and this method provides an alternative model for an animal tumor model and 2D culture to study cancer biology [138]. In another study, a tissue-specific micro-environment was developed using gastric tissue-specific bio-inks and cellulose nanoparticles and gastric dECM. The study showed that using cellulose nanoparticles increased mechanical properties, thereby promoting gastric cell aggressiveness, and it could be an effective model to develop for understanding gastric cancer biology [139]. The effect of Synthetic β-tricalcium-phosphate structures on the association between neuroblastoma tumor cells and stromal components was studied and found that the tumor micro-environment was shaped by stroma and maintained the growth of neuroblastoma cells. Additionally, cytokine and fibronectin production were stimulated, and the data provide information on how the 3D micro-environment stimulates tumor cells to form the spheroid shape and helps in understanding metastatic neuroblastoma [140]. In another study, the MCF-7 cell growth micro-environment was mimicked via 3D printing technology using Cs/Gel composite scaffolds. The scaffold developed exhibits good mechanical potential with enhanced biocompatibility and a precise platform for drug screening. The effect of Geniposide was tested in the 3D culture system and found that cell proliferation was inhibited with increased cell apoptosis and suggesting the anticancer effect of Geniposide [141].

### 4.3. 3D Printing and In Vitro Cancer Models

Generally, 2D models are used to understand cancer biology and conduct drug discoveries. However, 2D models do not replicate the in vivo tumor micro-environment, and animal models are also costly and have limitations due to species differences [142]. In this regard, a low-cost 3D printed model that resembles an in vivo system is receiving a lot of interest, and, in this section, we will be discussing how 3D printing techniques help to develop in vitro models. In a study, a patient-specific 3D printed silicon model of a stenotic aortic valve model was developed and kept inside a hemodynamic model of the blood circulation system to understand the flow of blood in the heart that would simulate patient-specific parameters. Data showed that 3D printed aortic valves effectively simulated hemodynamics and pressure gradients, which were predicted accurately. However, the study concluded that further refinement of the model and the addition of calcification are required to improve [143]. A study has fabricated a triple-layered human alveolar lung model consisting of fibroblast, endothelial, and lung epithelial cells using DOD 3D printing technology. The model helps to maintain cell viability and proliferation for a long period of time when related to non-printed cells, and thus this model can act as an efficient human alveolar in vitro lung model in the future [144]. A 3D printed polymeric film was fabricated with the topical delivery of berberine for the treatment of skin diseases using stereolithography with PEGDMA as photopolymerizing resin and PEG 400 would help berberine solubility and permeability. A drug permeation study revealed that the ex vivo diffusion rate of berberine was much higher and the in vivo skin irritability study showed that 3D printed films are non-irritated in nature, and thus this film could be effectively used to study for the treatment of skin diseases and drug screening in the near future [145]. A cell laden scaffold was fabricated using modified Ink H4-RGD and non-small-cell lung cancer (NSCLC) patient-derived xenograft cells were grown, which displayed increased viability and fast spheroid growth of the tumor size micro-environment formation within seven days. Three-dimensional spheroids are highly resistant to a drug like docetaxel, doxorubicin, and erlotinib when compared with 2D cultures, and thus the study suggested that it could be an effective in vitro model to analyze various drug screenings for cancer therapy [146]. A comparative study was conducted between a hybrid scaffold (polycaprolactone –hydrogel) and a single polycaprolactone scaffold based on the degradation nature and tissue compatibility in vitro and in vivo (Merino–Dorset sheep). The degradation volume was much higher in a single scaffold when compared to a hybrid and the 6-month study showed that the skin irritation or infection study showed that the polycaprolactone–hydrogel scaffold showed no skin irritation and exhibited an increased level of tissue in-growth [147]. An osteo-promoting 3D scaffold was developed using a 3D printed polycaprolactone scaffold with polydopamine and an osteo-promoting specific peptide (bone formation peptide-1). Later, in 3D printed scaffolds, human tonsil-derived mesenchymal stem cells (hTMSCs) were grown, which were differentiated into osteoblast-like cells. Additionally, the scaffold implanted in the calvarial defect rabbit model showed that bone and vessel regeneration was observed and suggested that this scaffold could be used efficiently in bone regenerative medicine [148]. In another study, a hybrid scaffold was developed using a decellularized bone matrix together with polycaprolactone, which exhibited increased mechanical properties and was easily printable. Later, culturing the scaffold with ADSCs exhibited an osteogenic transition with the expression of osteogenic genes. Adipose stem cell-seeded scaffolds were effectively entrenched into mouse calvarial faults, and it was found that the hybrid scaffold showed bone generation when compared to single polycaprolactone after three months of transplantation, and suggested that it could be a potential model for stimulating bone regeneration [149]. Mahmoudifar and Doran [150] reported that human fetal chondrocytes were implanted in the PGA scaffolds and cultured in the column bioreactors, which are combined with a perfusion system containing a magnetic stirrer. This culture method showed the collagen concentrations than the cartilage cultures. Yuste et al. [151] extensively reviewed 2D and 3D in vitro models, cell culture conditions, bone micro-environment, and the challenges and limitations.

### 4.4. Cancer Drug Delivery/Screening and 3D Printing

As cancer metastasis and recurrence pose a serious threat to the life of cancer patients, it is difficult for a chemotherapy strategy to succeed with a healing concentration of the drug at the cancer site. Several drug delivery systems are used to overcome the hurdles of systemic delivery; however, the necessary concentration of drug cannot reach the site due to solubility issues and the dissolution of the drug inside the carrier [152]. Considering all the limitations, 3D printing technology provides the possible solution for drug delivery and drug screening, which will be discussed in this section. A study has developed a 3D printed biodegradable patch for the release of a therapeutic drug concentration at the site of a tumor in a controlled manner. The patch consisted of 5-fluorouracil, polycaprolactone, and poly(lactide-co-glycolide) and was flexible with the capacity of releasing the drug over four weeks and decreasing the cancer cell growth in the pancreatic cancer xenograft mice model. The study suggested that it might be a prevailing strategy for the effective delivery of chemotherapeutic drugs for cancer therapy [152]. The 3D printing of drug nanocrystals was done using a method of semi-solid extrusion. Hydroxypropyl methylcellulose combined with glycerol as plasticizers were the film-forming polymer. The method optimized that the effective mechanical film potentials were obtained at two concentrations of the polymer, 3.5% (*w*/*w*) and (2.85% (*w*/*w*), and the particle size of the drug nanocrystal was found to be 230 nm. Ultimately, the study concluded that drug nanocrystal 3D printing via oral polymeric film formulations is an effective method for instant drug release with high solubility [153].

A 3D printed microswimmer with cargo loading and swimming potential was developed using gelatin methacryloyl and functionalized nanoparticles, which were made up of superparamagnetic iron oxide. Under a usual functional concentration, matrix metalloproteinase-2 would degrade microswimmer within 118 h; however, during pathological conditions, a microswimmer would respond fast to the matrix metalloproteinase-2 concentration by swelling and releasing the drug molecule. Additionally, upon the degradation of the microswimmer, it would release other cargos like anti-ErbB 2 antibody-tagged magnetic nanoparticles to label breast cancer cells in vitro, thereby helping in the medical imaging of remaining cancer tissues [154]. E-jet 3D printing was used to make poly-lactic-co-glycolic acid scaffolds that released doxorubicin and cisplatin into breast cancer cells, which caused the cells to die and stopped the growth of the tumor. This method could be used for a long time to deliver multiple drugs and prevent tumor recurrence [155]. A 3D printed calcium phosphate cement scaffold was used to deliver the anticancer drug 5-fluorouracil and was coated with a coating solution of hydrophilic Soluplus and polyethylene glycol. The in vitro data showed that the cell growth of the Hek293T-human kidney immortalized cell line and HeLa-human bone osteosarcoma epithelial cell line were decreased after 5 days. The study suggested that it could be successfully used to treat bone cancer and as a personalized medical solution for tissue engineering applications [156]. Three-dimensional-printed nanogel disc rounds act as drug carriers in delivering paclitaxel and rapamycin via intraperitoneal administration in ES-2-luc ovarian-cancer-bearing xenograft mice and are reported to be therapeutically effective in preventing postsurgical peritoneal adhesion in cancer-bearing mice [157]. A novel drug delivery device was developed using an alginate shell and poly (lactic-co-glycolic acid) core with the release of fluorescent dyes and exhibited no cytotoxicity in the HEK cell line or bone marrow stromal stem cells (BMSSCs). The study reported that it is an effective strategy in the controlled release of drugs or proteins via the delivery device and could be used for treating various cancers [158]. An alginate-gelatin/polycaprolactone core/shell scaffold coated with polydopamine was 3D printed and near-infrared laser stimulated drug (Doxorubicin) release, which led to the inhibition of tumor growth in both the in vivo and in vitro conditions. Moreover, the scaffold helps in wound healing and is also suggested to be implanted in the site to kill recurrent cancer cells and to repair tissue injury caused by surgery [159]. Using wax printing, paper with patterns of culture area, hydrophilic channel, and barrier area can be easily fabricated, and this printed paper could be used for a tumor cell culture to perform drug screening. A cell viability assay and drug sensitivity analyses showed that this wick paper-based microfluidic device could be an effective strategy for analyzing the drug screening and antibody drug production with low cost and ease of operation [160]. A 3D printed scaffold using gelatin methacryloyl was fabricated and the cells were cultured. A 3D bladder cell culture showed a higher cell interaction by increasing the secretin of E-cadherin and N-cadherin. Comparatively, the effect of rapamycin tested in both 3D and 2D cultures showed that the cells were more exaggerated in a rapamycin-treated 2D culture when compared to a 3D culture, and the study suggested that it could be used as an effective cancer cell environment to study drug screening [161].

### 4.5. Drug-Eluting Implant and 3D Printing

The drug-eluting prolapse mats made up of 3D printed material are biodegradable; as well, they are fabricated using coaxial electrospinning techniques and extrusion 3D printing. To mimic the extracellular matrix, the mat was made of polycaprolactone mesh with metronidazole, estradiol, and lidocaine-incorporated poly(lactic-co-glycolic acid) nanofibers that mimic the structure of most connective tissue’s natural ECM. The study showed that a nanofiber mat helps in the controlled release of estradiol, lidocaine, and metronidazole for 30, 25, and 4 days, while CTGF was released for more than 30 days. Additionally, the animal data showed that the nanofibers did not induce any side effects and suggested that it can be effectively used for the treatment of pelvic organ prolapse repair [162]. The three-dimensional (3D) printing of polycaprolactone/nano-hydroxyapatite using an extrusion-based printer showed that drug-eluting PCL/nHA screws eluted increased stages of antimicrobial ceftazidime and vancomycin for 14 days [163]. The 3D printing of bioactive-laden bioabsorbable catheters (14-F shape) was performed using powdered gentamicin sulfate or methotrexate coated on polylactic acid pellets and was tested on bacterial broth and plate cultures. Three-dimensional catheter constructs exhibited the controlled release of gentamicin sulfate and methotrexate for up to 5 days and effectively inhibited the growth of bacteria, which suggested that this strategy could be used to create an instrument for percutaneous procedures [164]. Three-dimensional printed hormone-loaded meshes (thermoplastic polyurethane mesh) were developed using different concentrations of 17-β-estradiol and the hot-melt extrusion method. TPU meshes were observed to be more elastic and suitable for pelvic floor repair. Additionally, 3D printed meshes showed the controlled release of estradiol for two-week periods, and it can also be changed using different concentrations of the 3D printed safer mesh implant [165]. Bioresorbable nanofibrous drug-eluting cuboid frames were developed for alveolar bone repair by 3D printing and electrospinning methods. Three-dimensional printed frames consist of polylactide cages (ketorolac and amoxicillin-loaded PLGA frames) and are tested for the treatment of alveolar bone defects. There was a sustained release of the drug for over 4 weeks and an animal showed greater movement without any adverse effects in the drug-eluting cuboid frame implanted group. By making use of 3D printing and electrospinning methods, cuboid frames with drug eluting potential can be developed for other maxillofacial applications [166]. In a study, a 3D scaffold loaded with rifampicin using polycaprolactone was developed for the treatment of osteomyelitis. A scaffold with antibiotics exerted growth inhibition potential against *Escherichia coli* and *Staphylococcus aureus* and did not exhibit any damage to human osteoblast growth in a 3D scaffold, which suggested that it could be used for the treatment of osteomyelitis [167]. A study has developed a patient-specific stenting process using biodegradable polymer doped with graphene nanoplatelets composite and a dual drug incorporation capacity. Based on drug loading and release measures, this strategy is effective in placing a stent in the coronary artery of the swine [168]. A micro-scaffold cochlear electrode array and a 3D micro-scaffold coated with dexamethasone encapsulated in PLGA was implanted into guinea pigs and analyzed the acoustic response evoked in the auditory brain stem. The threshold shifted to be lower in the implanted group and suggested this strategy helps in the development of cochlear electrodes with improved hormone release dynamics [169].

### 4.6. Cancer Metastasis and 3D Printing

Cancer that starts spreading to the distant parts of the body is called metastatic cancer, which would affect the efficacy of chemotherapy or radiation therapy. Thus, understanding and targeting metastatic cancer would help to develop an effective treatment strategy and improve patients’ survival. A study has developed an intelligent 3D printed scaffold made up of PLGA, gelatin, and chitosan and is loaded with anticancer drugs, which exhibit excellent hemostatic effects and act as a good pH sensor. This scaffold was implanted in situ in wounds, and a study showed that scaffolds effectively absorb hemorrhages and cells that are caused by surgery, which ultimately stimulates wound healing. On the other hand, the drug will be released based on the difference in pH. In a tumor acidic environment, there would be a sustained release of drugs without causing damage to normal cells, thus inhibiting the recurrence, growth, and metastasis of tumors. This strategy could be an effective treatment modality by providing excellent breast cancer therapy [170]. A 3D printed nanocomposite matrix has been developed using a stereolithography-based 3D printer and with a nano-ink consisting of hydroxyapatite NPs in hydrogel to mimic a bone-specific environment for assessing BC bone invasion. Additionally, a BC cell culture in a 3D printed matrix developed a spheroid phenotype and migratory potential, and, additionally, co-culturing with BMMSCs showed the enhanced formation of spheroid clusters and exhibited drug resistance potential. Thus, it has been suggested that the matrix could be an excellent tool to study metastasis and evaluate drug sensitivity [171]. A 3D printed bone scaffold was developed and conditioned with osteoblast-like cells, collagen matrix, and calcium. This scaffold was then cultured with patient-derived metastatic breast cancer cells and the data showed an increased survival of cells in the bone model and modulated drug response [172]. The 3D scaffold was printed using E-jet 3D printing to mimic the tumor micro-environment and p53-deleted cancer cells were cultured on the 3D scaffolds. Upon p53 deletion, cancer cell migration and proliferation decreased rapidly, thereby reducing cancer metastasis; thus, it could be an effective strategy for conducting tumor metastasis research [173]. Similarly, 3D-printed scaffolds loaded with DOX, which acts as a bone substitute, were cultured with a PC cell line and patient-derived spine metastases cells. Upon the release of DOX, the metabolic activity proliferation was reduced in both cells and thereby inhibited the metastasis [174]. Three-dimensional printed individual template-guided 125I seed implantation was done based on contrast-enhanced computed tomography images for the therapy of cervical lymph node metastasis. After surgery, no complications developed, and the implantation lowered the difficulty of puncture. This strategy was observed to be a safe and accurate guided approach [175]. Pang et al. [176] fabricated a HeLa/hydrogel grid consisting of gelatin, alginate, Matrigel, and HeLa cells using forced extrusion printing. Upon culturing cancer cells through the proliferation and attainment of spheroid structures with tumorigenic characteristics and the supplementation of TGF-β, the cancer cells were caused to disintegrate and alter their phenotype into spindle-shapes that show mesenchymal protein expression, including vimentin and N-cadherin, and a decreased epithelial protein expression of E-cadherin. Thus, 3D constructs help in developing a epithelial–mesenchymal transition (EMT) model to understand metastasis and, additionally, the use of disulfiram and EMT inhibitor effectively inhibited the EMT process; thus, it could also help in drug screening for tumor metastasis. Based on 3D printing, a prosthesis containing paclitaxel and doxorubicin microspheres (PPDM) was fabricated with an average particle size of 3.1 µm and 2.2 µm. The microspheres effectively suppressed breast cancer recurrence and metastasis in the xenograft mice model [177]. In a study, a 3D printed liver model was developed using patients’ CT images, which were transformed into stereolithographic files, printed using a desktop 3D printer, and were assembled and filled with silicone. Ultimately, they have developed a liver model with visible vessels and colorectal metastasis at low cost (under $150), and thereby increased the accessibility of the 3D model for planning surgery, which would reduce operation times [178].

### 4.7. Cancer Diagnosis and 3D Printing

A 3D printed electrochemical sensor was developed using tumor marker CD133, which was found in LC cells. The surface of the sensor was coated with recrystallized recombinant S-layer fusion protein, which immobilizes CD133. The sensor consists of a ceramic substrate with noble metals for the sensing element and 3D-printed capillary channels to guide the clinical cancer sample of rapid detecting ability at a low cost [179]. A 3D surface microarray was developed using a benzoboric acid-modified gold-plated polymeric substrate to detect circulating tumor cells. Comparatively, 3D micro-assays showed a higher capture efficiency of circulating tumor cells than that of a smooth surface, and this surface is highly sensitive with low cost and could be a promising strategy for the diagnosis of an early stage of cancer [180]. A 3D-printed unibody immunoassay was used for measuring the chemiluminescence output from PC biomarker proteins, including prostate specific antigen and platelet factor 4, with detection limits of 0.5 pg/mL and a 30-min assay time. The device consists of three reagent reservoirs, a 3D network for passive mixing, as well as an optically transparent detection chamber with a glass capture antibody array for measuring the chemiluminescence with a CCD camera [181]. A bipolar electrode system with 3D printed microchannels was developed to minimize the clinical sample requirement. The anode pole of the bipolar was modified with a nucleolin AS1411 aptamer and treated with a secondary aptamer modified with Au NPs to increase the sensitivity and selectivity; this strategy is low cost and can detect with a limit of about 10 cells [182]. In another study, a 3D printed immunomagnetic concentrator was used to enhance the ECL detection of the circulating tumor cells in the blood, which is an indicator of metastatic progression and relapse. This 3D printed concentrator allows the cancer cells to get concentrated up to 100 times, thereby allowing the ATP luminescence assay to detect ten cells in the blood, which is ten times more sensitive than existing marketable kits. Thus, the 3D printed concentrator helps in enhancing the detection limit of the ATP luminescence assay for detecting circulating tumor cells [183]. Similarly, a 3D-printed microfluidic array was developed to detect multiple proteins with a very low detection limit. It employs ECL detection measures with a CCD camera, touch screen, and reservoirs to deliver samples and reagents to a paper-thin pyrolytic graphite microwell detection chip to complete sandwich immunoassays. This low-cost, miniature immunoassay is used to detect eight protein prostate cancer (PC) biomarkers in human serum samples within 25 min [184]. In a study, a 3D printed microfluidic device was developed and functionalized with anti-epithelial cell adhesion molecules to separate circulating tumor cells from the blood. The study also tested three breast cancer cell lines (BC), colon cancer (CC), and PC cells and found that the capture efficiency was more than 90% and the breast cancer cells were isolated from blood samples, which could be used for cancer diagnosis [185]. Similarly, in another study, circulating cancer cells are detected by integrating a 3D printed off-chip multisource reagent platform, a bubble retainer, and a single circulating cancer cell capture microchip, and CTC was identified within 90 min. Additionally, circulating tumor cells were measured in the blood of 19 different cancer patients and detected and compared with clinical data. The study suggested that it could be used for early screening and real-time monitoring for hepatocellular carcinoma with low cost, user-friendly, and automation [186]. A 3D printed supercapacitor-powered ECL protein immunoassay was fabricated using a 3D printer and used to detect three cancer biomarker protein prostate specific antigens, prostate specific membrane antigens, and platelet factor-4 in serum, which were captured on antibody-coated carbon sensors. The detection limits were observed to be 300–500 fg/mL and, additionally, measuring six prostate cancer patient serum samples showed a respectable correlation with the conventional single protein ELISA method [187]. A 3D-printed biosensor with PDMSreservoir was developed with fluorescence detection, consisting of a nanomaghemite core with a gold nanoparticle shell for the magnetic separation of metallothionein. Upon the quantification of the metallothionein cell lines derived from spinocellular carcinoma and fibroblasts, the values gathered were 90 nM in tumor cells and 37 nM in fibroblasts, which suggested that the sensor was able to work with low volumes (<100 μL), low costs, and high portability [188]. A 3D printed flow cell with the functionalized electrochemical sensor was developed for the rapid detection of hepatic oval cells, which expressed the OV6 marker on their membrane. To immobilize the OV6 antibody, multiwall carbon nanotube (MWCN) electrodes with a chitosan film served as a scaffold. The developed sensor is entrenched into the 3D printed flow cell to allow cells to be exposed to the functionalized surface and cyclic voltammetry and square wave voltammetry were performed to understand the efficiency and selectivity of the printed sensor, which suggested that it is a valuable device for cancer diagnosis and detection [189].

### 4.8. Cancer-On-A-Chip and 3D Printing

Cancer is a complex three-dimensional tissue that has the dynamic potential to crosstalk with neighboring tissues via various signaling pathways. Thus, a recreation of cancer tissue using a 3D culture system is more likely to recapitulate the cancer architecture when compared to a 2D culture system. However, a 3D culture has limitations in recreating the dynamics of the tumor niche. In this condition, Cancer-on-a-chip is a microfluidic device that recreates tumor physiology and allows for a continuous supply of nutrients or therapeutic compounds [190]. In this section, we will be discussing the recently developed Cancer-on-a-chip model. A study has developed a 3D cell printing method to develop a hypoxic cancer Cancer-on-a-chip using a computer simulation of oxygen distribution. Bio-inks containing glioblastoma cells and endothelial cells are used to print cancer–stroma concentric rings to recapitulate solid cancer. The chip would be able to induce hypoxia and stimulate malignancy with the expression of cancer markers. Thus, this chip helps to create a solid-cancer-mimetic microphysiology to bridge the gap between the in vitro and in vivo models for cancer research [191]. A 3D-printed polymeric lab-on-a-chip was developed to tune the intrinsic functionalities by using DLP technology. Acrylic acid was added to the photocurable formulation to expose carboxyl groups in the polymeric matrix without the need for functionalization steps. The chip was fabricated to detect angiogenesis markers, including vascular endothelial growth factor and angiopoietin-2, by immunoassay with a detection limit of 11 ng/mL and 0.8 ng/mL, respectively [192]. Metastasis is an important factor that leads to a deprived prognosis in cancer patients. A study has developed a metastasis-on-a-chip by co-culturing kidney cancer cells (Caki-1) and hepatocytes decellularized with liver matrix (DLM)/gelatin methacryloyl (GelMA)-based biomimetic liver microtissue in a microfluidic device that mimics kidney cancer cells that are metastasized to the liver for predicting treatment efficacy [193]. A hybrid hydrogel with GelMA and hydrolyzed collagen was fabricated with a well-ordered homogenous microstructure to model the tumor micro-environment and possesses a good permeability and adjustable mechanical stiffness. The invasion of breast cancer and lung cancer (LC) cells in hydrogel were compared with non-invasive breast and colon cancer cells and the data suggested that hydrogel is effective in forming a 3D tumor culture, and its potential would replace Matrigel in cancer invasiveness evaluations. Furthemore, it was applied in a Tumor-on-a-Chip system with 3D-bioprinting [194].

Spheroid-on-a-chip was developed by the retention of spheroids in semi-circular traps within the microfluidic device to study the penetration of nanoparticles into the tumor. The chip is composed of triple layers of PDMS with four culture chambers with semi-circular weirs with apertures to allow for perfusion flow that trapped HepG2 multicellular spheroids [195]. Breast-cancer-on-a chip was developed with a microvessel wall, ECM, and tumor spheroids to evaluate a carbon dots drug delivery system and suggested that it could be an efficient platform to provide a more accurate and low-cost in vitro model for fast drug screening [196]. In another study, liver-on-a-chip was developed with three-dimensional human HepG2/C3A spheroids to analyze drug toxicity assessment. Additionally, a bioreactor design was allowed to monitor the culture environment and it could be interfaced with a bioprinter to fabricate 3D hepatic spheroid constructs encapsulated within photo-cross-linkable gelatin methacryloyl hydrogel. The secretion rates of transferrin, albumin, α-1 antitrypsin, and ceruloplasmin were monitored in the construct. Treatment with 15 mM acetaminophen stimulated toxicity in the ahepatic construct, which was similar to in vivo and in vitro models [197]. An active fluidic device was fabricated using a benchtop 3D printer with two procedures that are like 3D printing and polishing, and has valve control capacities to control the nutrient and ligand delivery flow that leads to generation signals mimicking environmental stimuli—therapeutic screening can also be performed in a liver tumor spheroid [198]. In some cases, the bioavailability of oral drugs is low due to first-pass metabolism, which is an important obstacle in drug development. A study has developed a microfluidic chip using a 3D printer after CAD design to recapitulate the first-pass metabolism and maintain the Organoid- or Spheroid-on-the-chip. The effect of first-pass metabolism was evaluated using docetaxel. In the chip without a small intestinal organoid, the viability of colorectal adenocarcinoma spheroids was reduced due to drug efficacy. Alternatively, the chip with the small intestinal organoid showed no change in viability due to the first-pass metabolism. The study suggested that a microfluidic chip is a rapid and low-cost system to analyze the efficacy of a drug on the first-pass metabolism [199]. To study the nanomedicine transport dynamics, an artificial microvessel-on-a-chip was fabricated using 3D printing and comprised of microchannels similar to the diameter of tumor capillaries and a semicircular geometry. Human endothelial cells were seeded into the round-shaped channels to create artificial blood microvessels and the microchip was connected by 3D-printed reservoirs to a pressure controller for fluidic control. Under physiological conditions, the dynamic interaction of nanoparticles with the artificial endothelium, internalization, and accumulation was analyzed in real-time using high-magnification fluorescence microscopy [200]. Muscle cell simulation using a muscle-on-a chip was developed by interconnecting with an electrochemical sensing system to measure inflammatory marker release (interleukin 6 and TNF-α), with a sensitivity in the range of ng mL^−1^ [201]. Elastin-like protein (ELP) engineered hydrogels were developed as bio-ink and directly dispensed onto endothelialized on-chip platforms. Neural progenitor cells and spheroid aggregates of BC cells were cultured, and their viability was found to be up to 14 days. The study suggested that combining ELP, 3D bioprinting techniques, and on-chip platforms could be used for the development of functional tissue models [202].

## 5. Nanomaterial, Cancer, and 3D Printing

Recently, nanomaterials like nanosheets, nanostructures, and nanotubes are receiving tremendous attention in the biomedical field, which could be used for tumor therapy and drug loading capacities. Additionally, 3D nanofiber scaffolds have attracted extensive attention in tissue regeneration, like bone and skin, due to their resemblance to their extracellular matrix structures. Interestingly, 3D scaffolds displayed bone and cartilage regeneration abilities. We discuss the role of nanomaterials in 3D printing and also in Table 3. A study has developed a biomimetic NP formulation of Cu(DDC) with a Stabilized Metal Ion Ligand complex. A 3D-printed microfluidic device is designed to improve the fabrication of metal–organic nanoparticles consisting of bovine serum albumin/Cu(DDC)2, which exhibited good physicochemical properties and exerted antitumor activities against breast cancer cells by inhibiting the growth of cancer cells, and thus could be used for cancer therapy [203]. A functional NP-enhanced nerve conduit consisting of gelatin-methacryloyl hydrogen with drug-loaded MPEG PCL was fabricated using 3D printing to promote peripheral nerve regeneration. Nanoparticles help in the controlled release of drugs (Hippo pathway inhibitor) to enhance nerve generation by increasing the proliferation and migration of Schwan cells and enhancing the expression of neurotrophic factor; thus, it could be an effective strategy to be used for clinical application in peripheral nerve repair [204]. A bifunctional scaffold 3D printed from nano-ink (metallic polydopamineFeMg-NPs) is fabricated to load and release the metal ions to exert chemo-dynamic therapy together with photothermal therapy, and, ultimately, eliminate bone-metastatic tumors. Additionally, the controlled release of osteo-inductive Mg^2+^ from the bony porous 3D scaffold stimulates the new bone formation in the bone defected area [205]. Nanomaterial-based uses of 3D printing and their mechanism in cancer treatment have been illustrated in Table 3.

## 6. Current Challenges and Future Perspectives

Three-dimensional printing technology is a developing area that provides models, plans for surgery, enhances drug release, and helps in drug screening with an increase in the success rate. Still, there are certain challenges to be solved to move closer to clinical applications, like the printing of functional tissues with high resolution, use of multicomponent ink, etc. The important drawbacks of the printing system involve the material characterization to enhance the printing, and cell viability should not be affected by the photocurable bio-ink and ultraviolet laser [213]. Another important limitation for printing is recapitulating the physiological, as well as the functionality, of cells needed to be considered [214]. Thus, the fabrication of the material that preserves cell viability and extensive characterization of the material, thereby optimizing the printing parameters, needs to be investigated extensively. A bionic scaffold is not similar to native tissues or cells and thus remains a challenge to improving bio-ink hydrogels [215].

Moreover, there is no ideal immune-competent mouse model to recapitulate the role of the immune system in disease progression, and the knowledge of understanding the early events of disease progression is poorly understood [216]. Thus, developing Metastasis-on-a-chip together with the immune model system can help to solve the above limitations. Furthermore, understanding the composition of the extracellular matrix will help to design future bio-inks. A 3D scaffold fabrication with improved biocompatibility will help with the differentiation and proliferation of stem cells [217]. Three-dimensional printing helps in planning surgery, reduces surgical time, and provides experience for physicians to prepare for complex surgeries. However, the 3D printed implant could not be altered during surgery in the case of a situation due to tissue loss, bone loss, etc. [181]. In those conditions, fabricating implants with multiple holes could be made for wiring and using screws. During bioprinting, the controlled delivery of stem cells is difficult due to the lack of robust techniques. Thus, planning for hybrid printing can allow multiple materials to support cells and provide the architecture for the cell growth/differentiation that could overcome these difficulties [218]. However, to study cancer cell invasion and metastasis, deep knowledge of the interaction between cancer cell stroma, blood vessels, and lymphatic vessels needs to be gained in order to mimic the complexity of cancer in a real tissue environment. Generally, 3D printed cancer models can be developed to integrate with multiple cell types to mimic the tumor environment, though still linking to different cell culture setups remains a challenge. Additionally, it is necessary to fabricate the tumor type and micro-environment-specific models to develop exact drug dose requirements, and to understand the metabolism and toxicity of the drug. Another important limitation includes sterilization, which is necessary for printing cells—should it be considered [219]. Several polymers that are used for fabrication have a limited choice of sterilization and, further, drug stability and the viability of cells should also be considered while using high-energy required printers. Recently, 4D printing has been recognized as a new emerging technique and is used in orthopedics to print smart orthopedic implants that could change their shape upon implantation in a patient with respect to time. This technology uses shape memory polymers, memory alloys, smart hydrogels, etc., to provide more advancements than 3D printing [147].

## 7. Conclusions

In the last decade, 3D printing technology has acted in the interaction between human physiology, synthetic biology, and biomaterials, and has played a major role in the biomedical field by fabricating complex in vitro model systems, identifying effective drugs and their toxicity, helping physicians with surgical planning for complex tissues, developing Tumor-on-a-chip to reduce time and cost compared to in vivo analysis, mimicking tumor micro-environment, and understanding cancer metastasis and early diagnosis cancer-related proteins. This review also helps to deepen the understanding of the most recent advances that are foremost to the development of printing methods and the design of bio-inks that enhance printing capacity, tunability, the viability of the cells, and the growth/differentiation of cells. Important obstacles to the widespread adoption of 3D printing for many healthcare applications remains, but ongoing material and printing improvements can address these issues and open the door to broader uses of extrusion-based 3D printing as a transformative technology. Three-dimensional printing is also still limited in terms of multi-material printing and resolution, necessitating extensive material characterization. Gaining knowledge of complex tissue structures will help to develop in vitro models and can be used for drug development and personalized medicine, which provides a significant platform to study cancer pathology. Due to the rapid development and low cost of printers and sensors, it provides a better future for the 3D-printed diagnostic device to analyze biomarkers of different cancers and its associated diseases.

## Figures and Tables

**Figure 1 pharmaceuticals-15-00678-f001:**
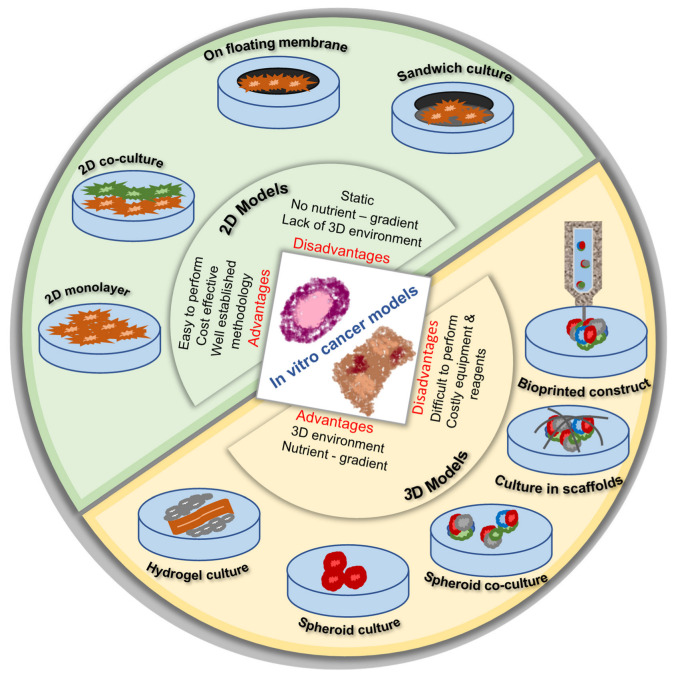
Various in vitro cancer models used in chemotherapeutic screening. Evolution of cell-culture models from simple 2D to complex 3D bio-printed models. Conventional 2D monolayer culture, monolayer co-culture, cells grown over floating membranes, and cell monolayer sandwiched between membranes, are the commonly used 2D cancer models in research and drug screening. Cancer cells cultured in hydrogels, spheroid monoculture, spheroid co-culture, cancer/stromal cells cultured in porous 3D scaffolds, and advanced bioprinted constructs are amongst the available 3D cancer models (Reprinted with permission from [17]).

**Figure 2 pharmaceuticals-15-00678-f002:**
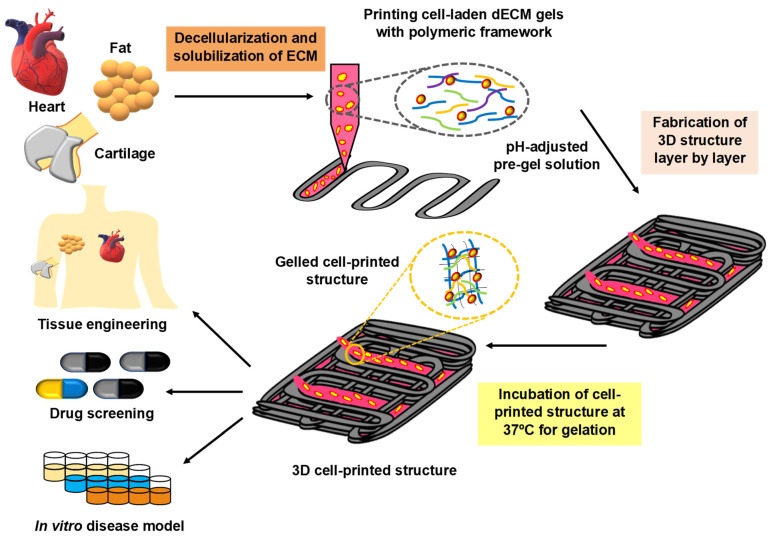
3D printing process and its application (Reprinted with permission from [18]).

**Figure 3 pharmaceuticals-15-00678-f003:**
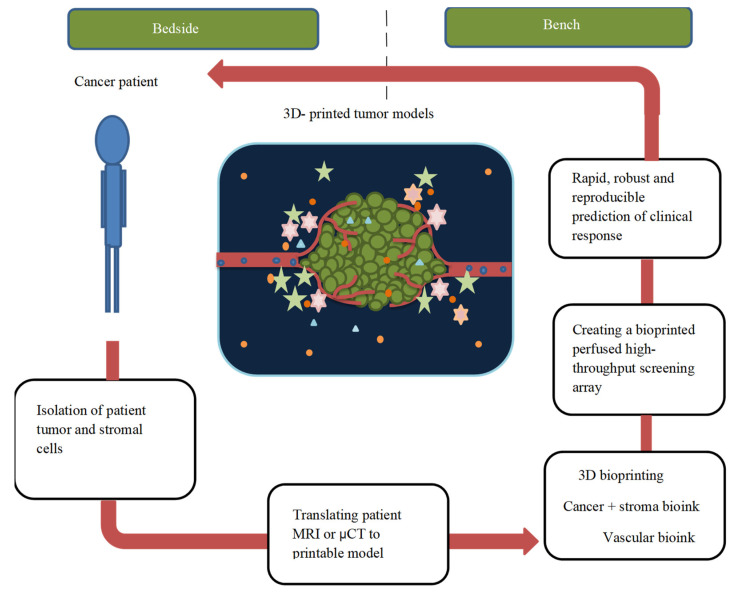
Schematic illustration of the methodological approach using a perfusable micro-engineered vascular 3D-bioprinted tumor model for drug screening and target discovery (Reprinted with permission from [30]).

**Figure 4 pharmaceuticals-15-00678-f004:**
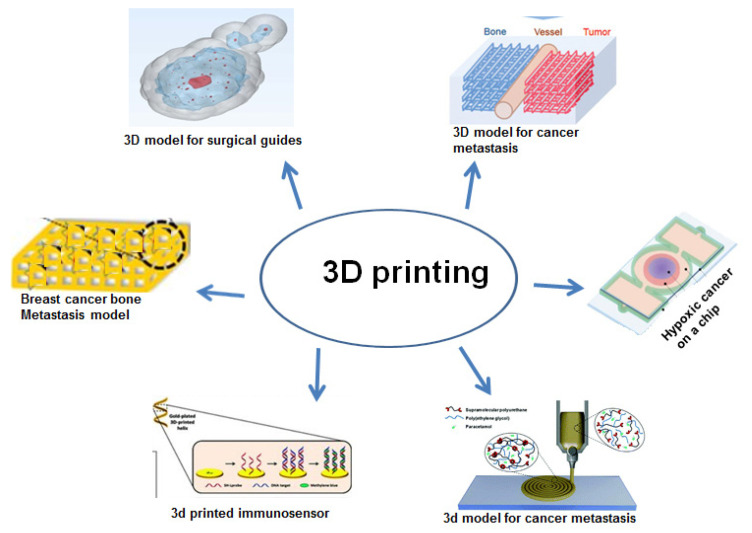
3D printing in cancer management.

**Figure 5 pharmaceuticals-15-00678-f005:**
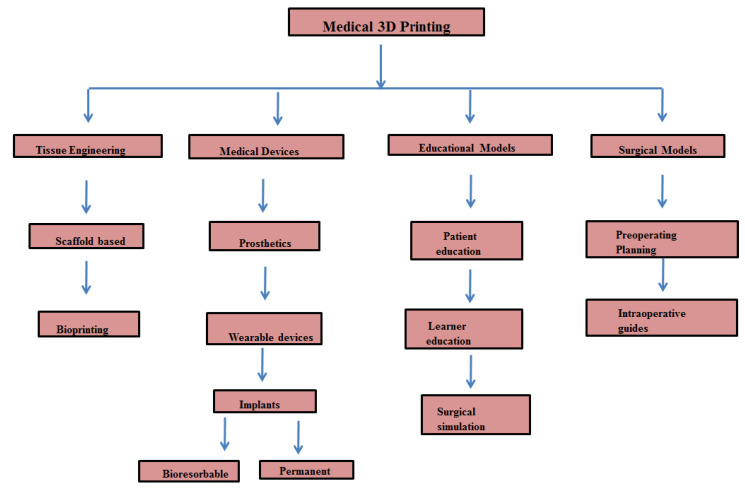
Classification of medical bioprinting and its applications (The figure was prepared based on the information provided by Anwar et al. [132]).

**Table 1 pharmaceuticals-15-00678-t001:** Mechanism and various biomedical applications of bio-ink.

S.No	Bioink Composition	Purpose	Mechanism	References
1.	Protein-photosensitizer conjugates.	Regenerative medicine	To fabricate protein gels cause cross-linking reaction on singlet oxygen. Submicrometer-scale precision. Improve the efficiency of gelation by cytocompatibility.	[60]
2.	Composite bioink comprises sodium alginate and egg white, often known as albumen.	Tissue and organ engineering	Sustain high viability. Vascular sprout and the development of a neovascular network.	[61]
3.	Composite bioink based on collagen/bioceramics.	Bone tissue regeneration	To develop a three-dimensional porous cell-laden composite material. Excellent cell viability and cell proliferation/differentiation. Exhibited significant osteogenic activities.	[62]
4.	The production of bio-ink from cell sheets.	To aid in the creation of various 3D geometries via bioprinting	An increase in the structural integrity. Reproducibility and automated deposition. Bioprinted constructions produced collagen type I, indicating that ECM deposition had started.	[63]
5.	Bioink that self-assembles and thins under shear (Methacrylated xanthan gum with gelatin bioink).	Creating bio-functional bioink for 3D bioprinting application	Supported cell viability during extrusion. Enhancement of nutrient/oxygen transport and cell motility.	[64]
6.	Composite hydrogel bioink with dual-cure (thermal/photo).	In situ 3D bioprinting	Demonstrated a quick thermo-induced sol-gel shift. Mechanical features that are adjustable. Proper microstructure and biodegradability	[65]
7.	Bioink with tunable Microgel-Templated Porogel (MTP).	To improve the use of 3D bioprinting.	MTP bioinks promote an increased metabolism rate. When seeding osteoblast cells, mineral production is more homogeneous than in bulk gel controls.	[66]
8.	Modular bioink: gelatin methacryloyl (GelMA)/chitosan microspheres	Nerve tissue engineering	Establishing an appropriate 3D microenvironment to promote neurite development. Establishing a 3D macroenvironment for Schwann cell proliferation and nerve cell organization by replicating the epineurium layer.	[67]
9.	Nanocomposite bioink	To produce tissue and organ surrogates for clinical use.	High viability of stem cells. Increases cell spreading and proliferation by boosting the rapid creation of extracellular matrix produced by cells.	[68]

**Table 2 pharmaceuticals-15-00678-t002:** 3D printed biosensors and their applications.

S.No	Biosensor	Application	Mechanism	Reference
1.	Microfluidic paper-based analytical devices	Using tiny nucleotide sequence changes to distinguish dengue virus serotypes	3D-printed barrier paper and a fluidic chip are combined.	[109]
2.	3D-printed nanocarbon electrode based on glucose oxidase	Detection of glucose in samples	To enable biosensing, a covalent linking approach was used to an enzyme on the surface of a 3D-printed electrode.	[110]
3.	Enzyme biosensor	Detection of hydrogen peroxide	Direct electron transfer enzyme-based biosensors are built using 3D-printed graphene/polylactic electrodes and horseradish peroxidase immobilization.	[111]
4.	Non-invasive 3Dprinted biosensor	Detect electrophysiological information	Sensor can measure electroencephalogram and electrocardiogram from zebrafish	[112]
5.	3D printed Chiral biosensor	Enantiomer recognition.	A 3D-printed electrochemical chiral sensor was functionalized with a magnetic covalent organic framework and BSA (chiral surface).	[113]
6.	Microfluidic reactor array manufactured in 3D	Molecular diagnosis of infectious disease	Isothermal amplification by Loop mediation in 50 min. The exposure limits for *Plasmodium falciparum* were 100 FG and 50 CFU for *Neisseria meningitidis* per treatment.	[114]
7.	Glucose dehydrogenase 3D printed glucose biosensor	To detect physiological glucose concentrations	As indicated by the slope and R2 correlation, a 3D-printed substance with a mylar substrate was immersed in an enzyme solution for 420 min.	[115]
8.	3D printed chemiluminiscencebiosensor	Lactate detection in oral fluid and sweat	3D printing technology is utilized to create a disposable small cartridge that could be readily prototyped to turn any smartphone or tablet or into a portable luminometer capable of detecting chemiluminescence resulting from an enzyme-coupled reaction with detection limits of 0.5 mmol/L.	[116]
9.	Nanomaterial enhanced 3D printed biosensor	Atrazine and acetochlor, two commonly used herbicides, were developed.	The catalyst of a mesoporous core-shell platium @palladium NPs on the redox reaction of thionin acetate and H_2_O_2_ produced an electrochemically driven signal that precisely showed the quantity of herbicide remains.	[117]

**Table 3 pharmaceuticals-15-00678-t003:** Nanomaterial based application of 3D printing and their mechanism in cancer treatment.

S.No	Nanomaterial	Disease	Mechanism	References
1.	Ultrathin copper-tetrakis (4-carboxyphenyl) porphyrin (Cu-TCPP) nanosheets interface-beta structured -tricalcium phosphate (TCP) scaffold	Bone tumor and bone defect	Assisted BMSCs and HUVEC connect. Increased osteogenesis differentiation-related gene expression angiogenesis and differentiation genes. Integration into rabbit bone defects stimulated bone repair.	[206]
2.	Muscle-inspired nanostructure: 3D-printed bioceramics scaffolds with a Ca-P/polydopaminenanolayer surface that self-assembles consistently.	Bone Cancer therapy and bone regeneration	Promote rabbit bone mesenchymal stem cellular proliferation. Even when photothermal therapy was used, the development of new bone tissues in rabbit bone defects increased.	[207]
3.	3D Printed WesselsiteNanosheets (Wesselsite [SrCuSi4 O10] nanosheets, SC NSs)	Vascularized bone regeneration	Extensive hyperthermia was caused by trigger osteosarcoma ablation with NIR-II light. Enhance cellular proliferation and osteogenic differentiation of rat bone marrow mesenchymal stem cells in vitro. Enhancement of vascularized bone regeneration.	[208]
4.	Tunneling nanotube (TNT) -like functional cell projections	Renal tumor microenvironment	The presence of 786-O renal carcinoma cells was due to cell viability and proliferation. Mitochondrial scrolling and intercellular transfer channels	[209]
5.	Cellulose nanofibrils (CNF), alginate, and SWCN are all examples of CNF-based materials.	Neuroblastoma	Neural cell differentiation In vitro 3D neural model to understand neurodegenerative disease	[210]
6.	3D printed materials containing cellulose nanocrystals (DS3000 and poly(ethylene glycol)diacrylate, PEG-DA) (CNCs).	Tumor microenvironment	Fine-tuning the nanostructure and functionalization of various 3D-printable substances.	[211]
7.	Polydopamine/Transferrin Hybrid (PDA/Tf) NPs	Cell killing	Melanoma cells treated with PDA/Tf nanoparticles experienced apoptosis after irradiation, which was mediated by lysosomal membrane permeabilization.	[212]

## Data Availability

Data sharing not applicable.

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
