# Peer review of "Three-Dimensional (3D) Printing in Cancer Therapy and Diagnostics: Current Status and Future Perspectives"

_pharmaceuticals, 2022, doi:10.3390/ph15060678_

Round 1

Reviewer 1 Report

Overall, the topic is very interesting for researchers as this is a hot topic and research is moving further very fast. So in my opinion, it is useful to have reviews that summarise the current information.  Only, there are few comments that should be addressed before publication. 

  • Fig 1 and 2 are very illustrative, congratulations. However, Fig 4 is not clear, resolution is poor. I am not sure if this figure is actually needed or it can be redrawn for clarity.
  • The are few spelling mistakes for example in line 635.
  • It would be interesting if authors can create other artwork to summarise info from other sections that are a bit more dense.
  • There are few points that should be cover a bit more into detailed for example: 4D printing in which time is the fourth factor. This is important when patients goes under breast mastectomy and the implant is reabsorbed over time. Check this ref: Applications of 3D printing in cancer
  • Also, there are other references that should be included for example, this one "Mimicking bone microenvironment: 2D and 3D in vitro models of human osteoblasts" that described in more detail the use of microfluidic chips to create a bone model.
  • Also, it will be interesting if you describe in more detail the materials neccessary for printing and which characteristics are important for their use when printing. Check this ref: Personalised 3D printed medicines: which techniques and polymers are more successful?

Author Response

Response to Reviewers’ Comments

Thank you for giving us the opportunity to submit a revised draft of the manuscript "Three-Dimensional (3D) Printing in Cancer Therapy and Di-agnostics: Current Status and Future Perspectives" for publication in the Pharmaceuticals. We appreciate the time and effort that you and the reviewers dedicated to providing feedback on our manuscript and are grateful for the insightful comments on and valuable improvements to our paper. We have incorporated most of the suggestions made by the reviewers. Those changes are highlighted within the manuscript. Please see below, in highlighted yellow, for a point-by-point response to the reviewers’ comments and concerns.

Reviewer 1 comments

Comments and Suggestions for Authors

Overall, the topic is very interesting for researchers as this is a hot topic and research is moving further very fast. So in my opinion, it is useful to have reviews that summarise the current information.  Only, there are few comments that should be addressed before publication.

Comment: Fig 1 and 2 are very illustrative, congratulations. However, Fig 4 is not clear, resolution is poor. I am not sure if this figure is actually needed or it can be redrawn for clarity.

Response: Thank you for your valuable and constructive suggestion. The resolution has been improved (See Fig 5).

Comment: There are few spelling mistakes for example in line 635.

Response: Thank you for your valuable comment. We have corrected.

Comment: It would be interesting if authors can create other artwork to summarize info from other sections that are a bit more dense.

Response: Thank you for your valuable comment. Yes, we have provided new figure (See Fig 4) .

Comment: There are few points that should be cover a bit more into detailed for example: 4D printing in which time is the fourth factor. This is important when patients goes under breast mastectomy and the implant is reabsorbed over time. Check this ref: Applications of 3D printing in cancer.

Response: Thank you for your valuable comment. As per reviewer suggestions, we have revised MS and application of 4D printing has been added in the revised MS. And new references

(Javaid, M., & Haleem, A. (2020). Significant advancements of 4D printing in the field of orthopaedics. Journal of clinical orthopaedics and trauma, 11(Suppl 4), S485–S490) has been  described in the revised MS (See line 1076-1081)

Comment: Also, there are other references that should be included for example, this one "Mimicking bone microenvironment: 2D and 3D in vitro models of human osteoblasts" that described in more detail the use of microfluidic chips to create a bone model.

Response: Thank you for your valuable comment. As per reviewer suggestions, we have revised MS and the reference  (Yuste I, Luciano FC, González-Burgos E, Lalatsa A, Serrano DR. Mimicking bone microenvironment: 2D and 3D in vitro models of human osteoblasts. Pharmacol Res. 2021 Jul;169:105626) has been  described in the revised MS (See lines 701-707)

Comment: Also, it will be interesting if you describe in more detail the materials necessary for printing and which characteristics are important for their use when printing. Check this ref: Personalized 3D printed medicines: which techniques and polymers are more successful?

Response: Thank you for your valuable comment. As per reviewer suggestions, we have revised MS (See lines 512-520).

Reviewer 2 Report

Dear Author

It is a very well written review on 3D-Bioprinting of tumor cells. One of the key challenge is rapid invivo analysis for translational applications. 3D-Bioprinting has the capacity to address this challenge. The topic is covered in depth and extensively with the applications of

Carbohydrates

Peptides

Biopolymers

Lipids

The microenvironment of the tumor is mimicked in utilizing this technique. This microenvironment is very well characterized and reported in several bioinks mentioned in the table. The table narrative is very impressive with mechanism of action in several possible commercially available inks.

Author Response

Thank you for giving us the opportunity to submit a revised draft of the manuscript "Three-Dimensional (3D) Printing in Cancer Therapy and Di-agnostics: Current Status and Future Perspectives" for publication in the Pharmaceuticals. We appreciate the time and effort that you and the reviewers dedicated to providing feedback on our manuscript and are grateful for the insightful comments on and valuable improvements to our paper. We have incorporated most of the suggestions made by the reviewers. Those changes are highlighted within the manuscript. Please see below, in highlighted yellow, for a point-by-point response to the reviewers’ comments and concerns.

Reviewer 2 comments
Comments and Suggestions for Authors

Dear Author
It is a very well written review on 3D-Bioprinting of tumor cells. One of the key challenge is rapid in vivo analysis for translational applications. 3D-Bioprinting has the capacity to address this challenge. The topic is covered in depth and extensively with the applications of Carbohydrates, Peptides, Biopolymers, Lipids. The microenvironment of the tumor is mimicked in utilizing this technique. This microenvironment is very well characterized and reported in several bioinks mentioned in the table. The table narrative is very impressive with mechanism of action in several possible commercially available inks.

Response: Thank you for your valuable, constructive and scientific appreciation. As per reviewer suggestions, we have revised MS.

This manuscript is a resubmission of an earlier submission. The following is a list of the peer review reports and author responses from that submission.